# Global and local tension measurements in biomimetic skeletal muscle tissues reveals early mechanical homeostasis

Arne D Hofemeier[1], Tamara Limon[1], Till Moritz Muenker[1], Bernhard Wallmeyer[1], Alejandro Jurado[1], Mohammad Ebrahim Afshar[2,3], Majid Ebrahimi[2,3], Roman Tsukanov[4], Nazar Oleksiievets[4], Jörg Enderlein[4,5], Penney M Gilbert[2,3,6], Timo Betz[1,4,5]†*

[1]Institute for Cell Biology, University of Münster, Münster, Germany; [2]Institute of Biomedical Engineering, University of Toronto, Toronto, Canada; [3]Donnelly Centre, University of Toronto, Toronto, Canada; [4]3rd Institute of Physics-Biophysics, University of Göttingen, Göttingen, Germany; [5]Cluster of Excellence "Multiscale Bioimaging: from Molecular Machines to Networks of Excitable Cells" (MBExC), University of Göttingen, Göttingen, Germany; [6]Department of Cell and Systems Biology, University of Toronto, Toronto, Canada

*For correspondence:
timo.betz@phys.uni-goettingen.de

Present address: †3rd Institute of Physics-Biophysics, Friedrich Hund Platz 1, University of Göttingen, Göttingen, Germany

**Abstract** Tension and mechanical properties of muscle tissue are tightly related to proper skeletal muscle function, which makes experimental access to the biomechanics of muscle tissue formation a key requirement to advance our understanding of muscle function and development. Recently developed elastic in vitro culture chambers allow for raising 3D muscle tissue under controlled conditions and to measure global tissue force generation. However, these chambers are inherently incompatible with high-resolution microscopy limiting their usability to global force measurements, and preventing the exploitation of modern fluorescence based investigation methods for live and dynamic measurements. Here, we present a new chamber design pairing global force measurements, quantified from post-deflection, with local tension measurements obtained from elastic hydrogel beads embedded in muscle tissue. High-resolution 3D video microscopy of engineered muscle formation, enabled by the new chamber, shows an early mechanical tissue homeostasis that remains stable in spite of continued myotube maturation.

## Introduction

Skeletal muscle is one of the most abundant tissues in the human body and is crucial for essential functions such as limb movement, thermogenesis, and maintaining posture (*Periasamy et al., 2017*; *Lauretani et al., 2003*; *Buckingham, 2001*). Skeletal muscle atrophy is debilitating and is a typical outcome of diseases like muscular dystrophies but also aging. Characterizing the biomechanics of muscle tissue is a key element to understand both, muscle development and muscle degeneration. Although this characterization has been largely achieved on the global tissue level, the local mechanical properties during muscle development and maturation remain inaccessible. This is largely due to the limited optical access of muscle tissue which prevents the use of modern fluorescence microscopy-based methods. Indeed, although molecular biology has contributed a phlethora of fluorescence microscopy-based tools to label molecules (*Heilemann et al., 2008*; *Walker et al., 2015*; *Bourgeois et al., 2012*), modify signaling cascades (*Fenno et al., 2011*) and measure molecular interactions (*Algar et al., 2019*) as well as mechanical tension (*Lee et al., 2019*; *Grashoff et al., 2010*; *Ringer et al., 2017*), the field of muscle tissue research is partially hampered in exploiting these tools due in part to the limited access to high end and super resolution fluorescence

microscopy. In recent years, the study of skeletal muscle development and disease is shifting from animal models to new in vitro muscle tissue approaches, that promise more controlled experimental approaches with improved optical access. Furthermore, these engineered muscle tissues not only avoid ethical considerations, but also overcome several problems of animal models such as a high price, time consuming procedures, and in some cases, failure to accurately predict human treatment response due to species-specific differences (*DiMasi et al., 2003*; *McGreevy et al., 2015*; *Gaschen et al., 1992*). Although 2D skeletal muscle cell cultures can provide ease of predictions during drug testing and disease modeling (*Stevenson et al., 2005*), these systems are of limited use in contraction studies due to randomly oriented myotubes and the inability to maintain long-term cultures (*Eberli et al., 2009*; *Smith et al., 2014*; *Guo et al., 2014*; *Pimentel et al., 2017*). In contrast, modern reconstituted 3D in vitro skeletal muscle systems are demonstrated to be an efficient tool for rapid and reliable drug screening (*Vandenburgh et al., 2008*; *Afshar Bakooshli et al., 2019*) and allow for testing of personalized treatments on cells harvested from individual patients. Besides these medical advantages, such functional muscle tissues were reported to successfully mimic native muscle tissue with long-term structural integrity (*Lee and Vandenburgh, 2013*; *Juhas et al., 2014*) and allow new insights and fundamental knowledge of muscle tissue development, force generation during contraction as well as the phases of disease onset and progression (*Madden et al., 2015*; *Maffioletti et al., 2018*; *Rao et al., 2018*; *Takahashi et al., 2018*; *Gholobova et al., 2018*; *Kim et al., 2018*; *Shima et al., 2018*; *Capel et al., 2019*; *Afshar Bakooshli et al., 2019*; *Kim et al., 2020*). In culture platforms allowing for in situ force measurements, 3D skeletal muscle tissues self organize between mm-sized posts around which muscle precursor cells are seeded together with an extracellular matrix. These chambers are commonly based on polydimethylsiloxane (PDMS) molds that can be tuned in their elastic properties and are soft enough to allow measurement of muscle contraction forces by recording the post-deflection. Such non-invasive measurements of force generation in reconstituted muscles are indeed based on the elastic properties of PDMS (*Legant et al., 2009*; *Afshar et al., 2020*). A downside of PDMS is that the poor optical properties paired with routinely used material thickness prohibits the use of high numerical aperture objectives that are necessary for high-resolution microscopy, and hence make modern 3D microscopy methods like confocal and spinning disk microscopy on living tissue impossible. Instead, to study 3D tissues with high resolution requires fixation and removal from the culture device, thus preventing research on questions related to dynamic processes. Hence, they are of only limited use for the investigation of spatiotemporal research questions such as dynamics of cell-cell interaction or myoblast fusion. Moreover, PDMS has an immense capacity to absorb chemicals and proteins and is therefore unsuitable for serum-free medium applications or precise drug evaluation (*Toepke and Beebe, 2006*).

Recently, a method to determine contraction forces of myotubes grown within collagen or other ECM-like matrices using 3D traction force microscopy was described (*Rausch et al., 2020*). This approach resolves force generation not only at the endpoints of the myotubes, as in the post systems, but can also determine traction forces along the length of the myotube. Although this new method dramatically enhances the resolution of force generation and is compatible with high-resolution microscopy, it is limited to forces acting on the interface between the muscle cells and the provided environment. The force and tension distribution within the tissue, and namely between individual cells, remains only accessible by locally cutting myotubes, and thus damaging the system.

While the local distribution of tension in reconstituted muscle tissues cannot be measured at the moment, mechanical tension and its distribution in other tissues was previously shown to guide many fundamental biological processes such as collective cell migration, tissue morphogenesis, and cell fate decisions (*Wallmeyer et al., 2018*; *Engler et al., 2006*; *Saha et al., 2008*). Despite this, non-destructive experimental approaches to investigate spatial and temporal forces on a cellular level are limited and thus, characterization of local cell niches within a tissue remains a challenge. To overcome this problem, Campas et al. introduced biocompatible oil microdroplets to evaluate cell-generated forces in living tissue for the first time (*Campàs et al., 2014*) and very recently, deformable (Polyacrylamide) PAA beads were used to determine local tension on a cell scale within cancer spheroids, zebrafish embryos and during phagocytosis (*Dolega et al., 2017*; *Lee et al., 2019*; *Träber et al., 2019*; *Vorselen et al., 2020*). Contrary to oil microdroplets, PAA beads are compressible and are therefore able to reveal isotropic tissue pressure.

In this study, we open new ways to study the global and local mechanical properties of biomimetic skeletal muscle tissue. We first present a novel technique to culture in vitro skeletal muscle

tissues in 3D with inverse geometry, so that the tissue is located closely to a microscopy glass slide. This allows real-time high-resolution imaging, while still self organizing around thin posts of polymethylmethacrylate (PMMA) that are connected to the top of the chamber and thereby allowing both, accurate quantification of global forces via post-deflection and high-resolution 3D fluorescence microscopy. Next, we incorporate custom-made elastic PAA beads into the biomimetic muscle tissues to trace local tension within the tissues during formation. Both, the global and local forces match in magnitude, as expected. Interestingly, we observe an immense increase in local tension inside muscle tissues in vitro within the first week of formation which remains at a similar level in the following week, although myotube diameter continues to increase. Notably, global muscle tissue pre-tension did not increase in the second week of differentiation, either. By enabling real-time high-resolution microscopy on 3D in vitro muscle tissues for the first time, we envision our novel reconstituted muscle tissue device will be an advantage to the skeletal muscle research community by enabling the dynamic study of (sub-)cellular events.

## Results

### In vitro formation of functional 3D skeletal muscle tissues compatible with high-resolution video microscopy

The combination of high-resolution microscopy and reconstituted three-dimensional (3D) skeletal muscle microtissues holds the potential to lend new insights into the formation and maturation of skeletal muscle cells. To enable such studies, we re-envisioned the design of currently used 3D muscle microtissue culture platforms, which led to the establishment of a two part chamber system. The bottom component is milled from a PMMA block to contain eight individual oval-shaped cell culture chambers (*Figure 1A*, bottom; *Figure 1—figure supplement 1*). This bottom part is glued to a standard microscopy coverslip, which then enables high- as well as super resolution inverted fluorescence microscopy compatible with oil or water immersion objectives. The upper part is also machined from PMMA and fits tightly into the bottom part. It consists of long vertical posts (*Figure 1A*, top; *Figure 1—figure supplement 1*) to which the 3D muscle tissue anchors during formation by wrapping around them. When the two halves of the culture device are placed together, the vertical posts nearly touch the bottom, thus confining the muscle tissue to a region close to the bottom of the coverslip. A hole in the top part positioned equidistant between each pair of posts allows for gas and medium exchange during growth and measurement (*Figure 1—figure supplement 1*).

Biomimetic 3D muscle tissues arise from a self-organization process when an initial mixture of mononucleated skeletal muscle progenitors (aka 'myoblasts') in a Geltrex-Fibrin matrix is seeded into the bottom of each device well (*Figure 1B*), similar to the tissue formation in previously described PDMS-based culture systems (*Madden et al., 2015*; *Afshar Bakooshli et al., 2019*; *Afshar et al., 2020*). To demonstrate the imaging quality and observation capacity of this chamber design, C2C12 mouse myoblasts were seeded within a Geltrex-Fibrin matrix at defined ratios and cultured in growth medium for 2 days allowing for equilibration to the 3D environment. The medium was then exchanged to a low serum formulation to support differentiation (i.e. fusion to form multinucleated 'myotubes') for a period of up to 2 weeks (*Figure 1C*). During remodeling, the tissues anchored around each end of the non-adhesive posts (*Figure 1B*). Resulting tissues fixed and immunostained for sarcomeric alpha actinin (SAA), and counterstained using Hoechst 33342 to visualize nuclei, revealed multinucleated myotubes aligned in parallel between the posts, as well as striations indicative of sarcomere structures characteristic of myotubes progressing through the process of maturation (*Figure 1D*). These in-plate observations of sarcomere structures required diffraction limited fluorescence microscopy, which is not possible in the context of a PDMS-based cultivation system, when imaging through the PDMS (*Figure 1E*). To further characterize biomimetic muscle tissue formation, images of transverse 3D muscle tissue cryosections were prepared that displayed myotubes which are consistently aligned close to each other and are evenly distributed throughout the entirety of the tissue (*Figure 1F*). The cross-section of the tissue was measured to be $0.17 \pm 0.03$ mm$^2$.

To demonstrate the capacity of time-lapse high-resolution microscopy on living 3D skeletal muscle tissues, we generated a Lifeact-GFP and H2B-mCherry-labeled human myoblast cell line

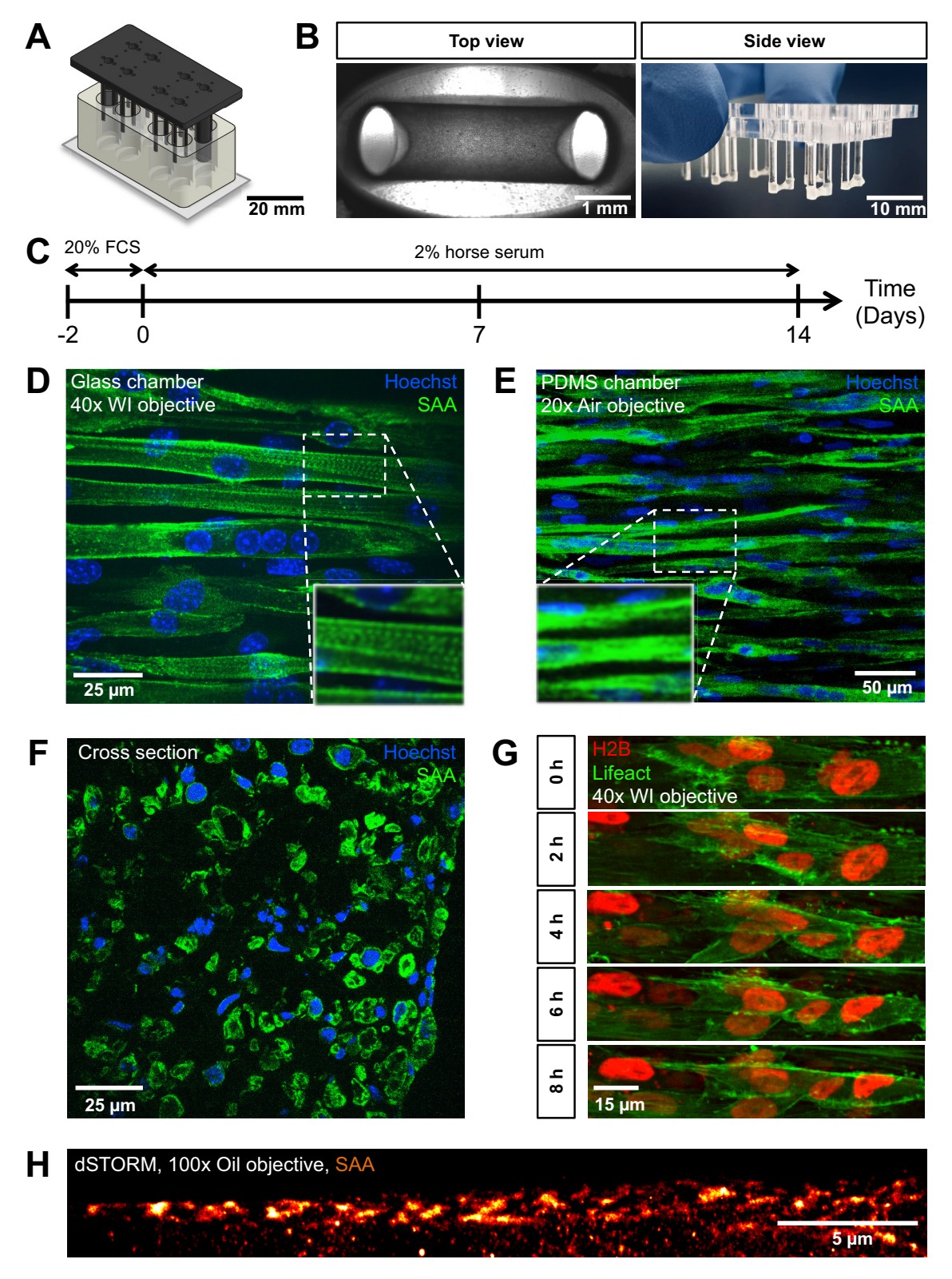

**Figure 1.** PMMA culture device supports the generation of 3D biomimetic skeletal muscle tissues. (**A**) Computer-generated depiction of the PMMA mold design. The top part containing eight pairs of vertical posts is shown in black. Eight holes positioned equidistant between each pair of posts allows for gas and media exchange. Two larger sized vertical posts at either end of the top part serve to fix the top and bottom portion (translucent white) together and ensure vertical posts are properly positioned. The bottom portion is affixed to a microscopy grade glass (**B**) Images of remodeled

*Figure 1 continued on next page*

*Figure 1 continued*

C2C12 muscle tissues at 7 days of differentiation anchored to the end of the posts captured to provide (left image) a view looking up into a well in which the top and bottom parts are fashioned together (left; ×4 objective) or (right image) from the side looking at the top part to visualize six pairs of posts with tissues at the bottom. (C) Schematic workflow used to raise 3D skeletal muscle tissues in vitro. (D–F) Representative confocal microscopy longitudinal (D; whole mount, flattened stack, ×40 water immersion objective, PMMA chamber, (E); whole mount, flattened stack, ×20 air objective, PDMS chamber) and transverse (F; cryosection, single snap, ×40 water immersion objective) images of multinucleated myotubes within a 14-day-old muscle tissue immunostained for sacromeric alpha-actinin (SAA, green) and counterstained with Hoechst 33342 to visualize nuclei (blue). (G) Timeseries of myoblast fusion within a 3D skeletal muscle tissue on day 4 of differentiation, demonstrating the possibility of high-resolution imaging during living tissue formation. Lifeact-GFP (green) and H2B-mCherry (red) was stably introduced into AB1167 cells. (H) Super-resolution dSTORM imaging of SAA-stained myofibrillar structures present inside a myotube, recorded with ×100 oil immersion objective.

The online version of this article includes the following figure supplement(s) for figure 1:

**Figure supplement 1.** Engineering drawing with respective dimensions of the novel culture device used in this study for raising 3D skeletal muscle tissues in vitro.

**Figure supplement 2.** Schematic of the custom-built setup for super-resolution dSTORM microscopy (A).

(AB1167-Lifeact-GFP-H2B-mCherry) and recorded a timeseries of several days old 3D skeletal muscle tissue dynamics using a spinning disk microscope. The results show that although the tissue appears immobile at the macroscopic level, or in snapshots, a continuous rearrangement at the cell level takes place. We observed myoblast fusion events to form myocytes as well as highly dynamic nuclear motion during progressive tissue formation (*Figure 1G*, *Video 1*). We further used our new chamber system to perform in-plate single-molecule localization-based super-resolution imaging on 2-week-old SAA-stained C2C12 tissues to emphasize the unlimited possibilities that are permitted through the cultivation of biomimetic muscle tissues on glass. Specifically, we used (direct) stochastic optical reconstruction microscopy (dSTORM) (*Heilemann et al., 2008*). The SAA staining revealed partly unaligned myofibrillar structures in a myotube that could be resolved in delicate detail (*Figure 1H*, *Figure 1—figure supplement 2*). Despite the fact that the signal-to-noise ratio is rather low due to the thickness of the sample, dSTORM with a high-NA objective was possible. We achieved super-resolution with a localization precision of 17 nm and an average resolution of 65 nm. These first super-resolution images of 3D muscle tissue were only possible by imaging through the glass bottom using a ×100 oil objective. Such highly technically challenging approaches are not feasible using devices which are mostly based on PDMS. Hence, our approach enables the formation of 3D biomimetic muscle tissues with characteristic skeletal muscle features that are in close proximity to a glass window, thus enabling time resolved 3D microscopy at the diffraction limit and even super-resolution imaging, which represents a highly advanced alternative to previously described culture systems.

## Global contraction forces of in vitro skeletal muscle tissues

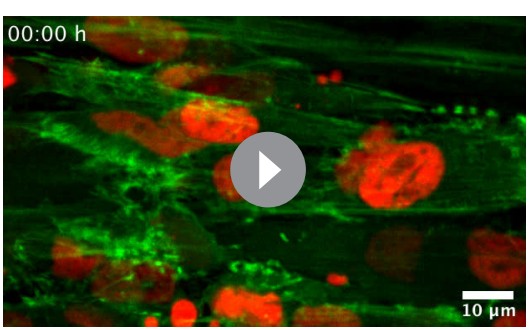

00:00 h

10 µm

**Video 1.** Flattened 8 hr timeseries stack of a developing human microtissue 4 days after differentiation shows highly dynamic nuclear motion and myoblast fusion events during myotube formation. Lifeact-GFP (green) and H2B-mCherry (red) was stably introduced into AB1167 cells.

https://elifesciences.org/articles/60145#video1

The most common approach to determine the contractile forces of a reconstituted muscle tissue is to monitor the deflection of elastic PDMS posts and to then calculate contractile forces using the spring constant of the posts (*Afshar et al., 2020*). Indeed, an advantage of PDMS is that the elasticity, and hence the spring constant, can be tuned by the degree of cross-linking during curing. Although the elasticity of the PMMA material used to construct our device cannot be easily changed, by varying the geometry of the posts, it is possible to adjust the spring constant. Namely, the diameter and the length of the posts are sensitive parameters responsible for their compliance. By adjusting the length of the posts, we can easily change the spring constant of the posts by more than one order of magnitude (*Figure 2F*). The spring constant of 16 mm long PMMA posts used in this

study was calculated to be 39 µN/µm, which was further confirmed experimentally by direct measurements to be 39.24 ± 0.78 µN/µm. Since the predicted and measured values for posts shorter than 12 mm start to diverge, we suggest careful spring constant measurements starting at this length to ensure reliable values. Although the stiffness of PMMA posts is considerably higher than commonly used PDMS-based posts, this disadvantage is compensated by the highly improved imaging capacity, that allows for determining deflection amplitudes down to 0.2 µm, which corresponds to a force resolution of ±7.8 µN.

To test if the global tissue contractile force detection is comparable to that reported for human skeletal muscle tissue formed in PDMS-based systems, 2-week-old skeletal muscle tissues generated from an immortalized human myoblast line (*Girardi et al., 2019*) genetically modified to stably express the light-sensitive channelrhodopsin-2 ion channel (*Boyden et al., 2005*; AB1190-Fubi-ChR2-GFP) were investigated in our chamber system. Consistent with prior work, we find a clear deflection of the posts upon comparing the position of the posts when the muscle tissue is relaxed versus in a contracted situation (*Sakar et al., 2012*). For both optogenetically and acetylcholine (ACh) triggered contractions, we measured deflections of several micrometers (*Figure 2A*, *Videos 2* and *3*). Owing to high-contrast imaging of the PMMA posts made feasible by the culture device, it is possible to precisely trace the post-deflection and determine the contractile forces. Our custom post-deflection analysis software is able to trace exerted forces on the post over time from acquired videos with minimal background signal of ±8.5 µN (*Figure 2B*). For optogenetic twitches, we observe a roughly 1-s lasting contraction phase before the in vitro muscle tissue returned to the relaxed state. By comparison, a tetanus contraction induced by 2 mM ACh reached its peak of contractile force more slowly and the time elapsed before return to the relaxed state was almost 25 s (*Figure 2C*). While optogenetically induced twitch contractions exhibited contractile forces of 0.2 ± 0.04 mN on average, the tetanus contractions induced by treatment with 2 mM of acetylcholine elicited contractile forces of about 1.1 ± 0.3 mN (*Figure 2D*). These measurements are consistent with force measurements obtained in PDMS-based chambers (*Afshar et al., 2020*).

We further established a method to evaluate pre-tension during skeletal muscle tissue formation in vitro. Specifically, we treated C2C12 skeletal muscle tissues with 10% SDS to dissolve the myotube membranes, thereby releasing the pre-tension established by the myotubes, which is visualized by the PMMA posts returning to their original position upon dissolution of the muscle tissue (*Video 4*). Utilizing our post-deflection analysis software (accessible on GitHub [*Muenker, 2020*]), we quantified the pre-tension to be 0.3 ± 0.1 mN for skeletal muscle tissues cultured for a period of 1 week (*Figure 2E*), and this did not change significantly in the subsequent week of culture. Taken together, we demonstrate that the culture device allows for determination of both contractile forces as well as tissue pre-tension simply via post-deflection analysis.

## Characterization and computational analysis of elastic PAA beads

To determine the local tension in the muscle tissue, we implemented elastic hydrogel beads as tension sensors, an approach first used by others in different systems (*Dolega et al., 2017*; *Lee et al., 2019*; *Träber et al., 2019*; *Vorselen et al., 2020*). We custom-made elastic PAA beads by a water in oil emulsion approach (*Figure 3A*). The use of PAA beads as tension sensors requires precise mechanical characterization. Using optical tweezer based active microrheology on 1 µm polystyrene particles embedded within the PAA beads, we determined a shear modulus of $G = 710 ± 270$ Pa by averaging shear storage moduli $G'$ for low frequencies from 1 Hz to 10 Hz (*Figure 3B*). Additionally, the bulk modulus was determined to be $K = 41 ± 6$ kPa by an osmotic pressure approach (*Figure 3C*). Here, three different concentrations of 2 MDa dextran (60, 85, and 100 g/l) were applied to the elastic beads, which corresponds to osmotic pressures of 6, 12, and 18 kPa, respectively. PAA bead diameter in

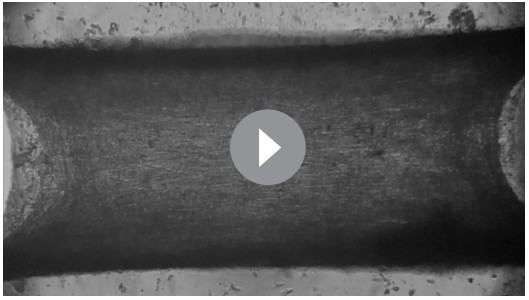

**Video 2.** Contraction of a human muscle microtissue upon optogenetically induced stimulation at two weeks of differentiation. Channelrhodopsin-2 was stably introduced into AB1190 cells.
https://elifesciences.org/articles/60145#video2

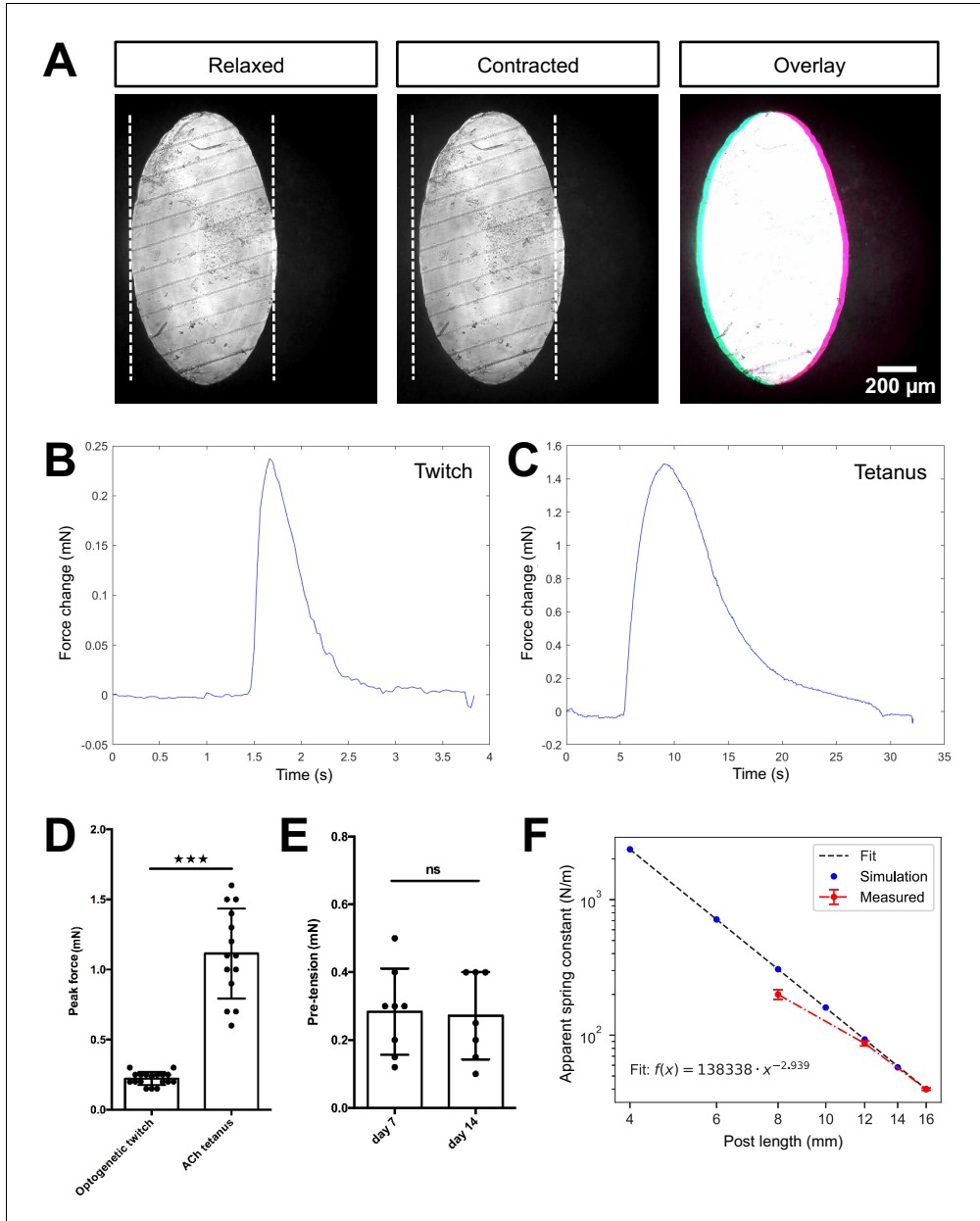

**Figure 2.** Quantification of biomimetic skeletal muscle tissue contractile forces over time in culture. (**A**) Representative bright-field images of the bottom of a post under ×10 magnification before (left) and during (middle) a tetanus contraction. Pink and blue pseudo color highlights the deflection regions when pre- and post-contraction images are overlaid (right). Representative time course traces of forces exerted on the posts are displayed in (**B**) for an optogenetically induced twitch and in (**C**) for a tetanus contraction induced by the addition of 2 mM ACh. (**D**) Bar graph with force measurements of optogenetically induced twitches (N = 18) and ACh induced tetanus contractions (N = 14) of 2-week-old muscle tissues. (**E**) Bar graph with pre-tension measurements of skeletal muscle tissues at 1 (N = 8) and 2 weeks of differentiation (N = 7). (**F**) Relation between simulated and measured spring constants for different post lengths. Up to a post length of 12 mm, the simulated and measured spring constant are equal within the error; however, for shorter posts we observe discrepancy. Hence, such short posts should be always crosschecked experimentally.

The online version of this article includes the following source data for figure 2:

**Source data 1.** Source data of global contraction and tissue pre-tension studies (*Figure 2B–E*).
**Source data 2.** Source data of spring constant determination of PMMA posts (*Figure 2F*).

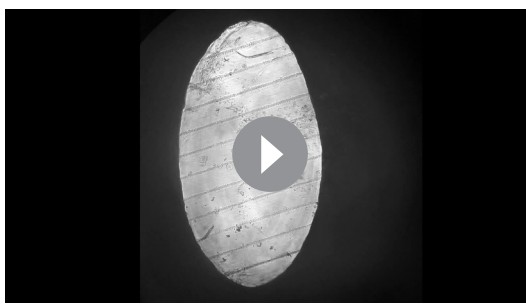

**Video 3.** Post-deflection upon contraction of a 2-week-old human AB1190 microtissue induced by 2 mM ACh. https://elifesciences.org/articles/60145#video3

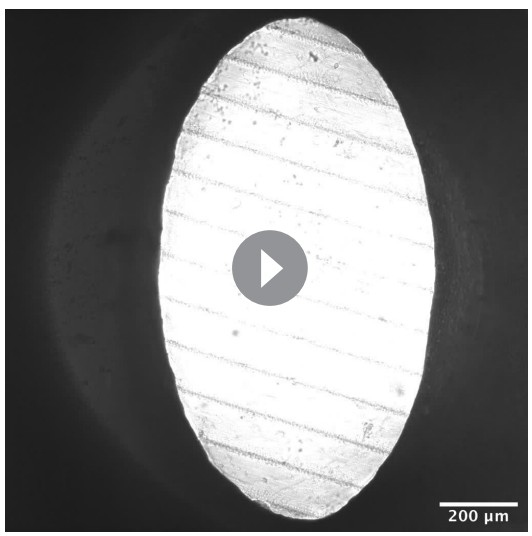

**Video 4.** Post-deflection during dissolution of a 1-week-old human AB1167 microtissue with 10% SDS. https://elifesciences.org/articles/60145#video4

the three solutions was measured and compared to the corresponding uncompressed situation. Consequently, the Poisson's ratio $\nu = (3K - 2G)/(6K + 2G)$ was determined as $\nu = 0.491 \pm 0.005$, which is close to incompressibility and consistent with previous reports of Poisson's ratio in PAA (*Boudou et al., 2006*). For the force calculation, the Young's modulus $E = 2G(1 + \nu)$ of the PAA beads used in this study was hence determined to be E = 2.1 ± 0.8 kPa with an average diameter of 8.4 ± 0.1 μm (mean ± SEM). The precise measurement of the Poisson's ratio is very important for the analysis, as the analytical solution diverges for a Poisson's ratio of exactly 1/2. However, as no real solid material exists with a Poisson's ratio of perfectly 0.5, this poses only a theoretical limit for the applied analysis. It should be noted, that for materials close to incompressibility, as the PAA used in this study, even a small error in the Poisson's ratio can have large influence on the resulting forces. This is a main reason for the herein employed measurement of two mechanical moduli that give direct access to the Poisson's ratio. The value of the Poisson's ratio is further confirmed by a crosscheck between the global and local forces as discussed later in the text.

We next used the mechanical properties of the PAA beads to determine the forces exerted on the beads, from images where bead deformation was measured. Previous studies by others used finite element methods for the analysis (*Träber et al., 2019*), which have the disadvantage of being computationally time consuming. Here, we focused on an analytical solution that yields the dominant tension dipole, and the spatial orientation of this dipole force. The advantage of this approach is the high speed of analysis, and the reduction of the relevant forces into a simple scalar force number, while reporting also the direction of force tension propagation in the tissue. We performed the analysis using a custom written program that implements an automated computational bead deformation analysis (BDA), which is capable of deconvolving microscope images using a given point spread function (PSF), extracting the edges of beads, and finally fitting spherical harmonics to the segmented beads to register the bead deformation (*Figure 3D*). Finally, BDA uses the identified coefficients of the spherical harmonic fit to analytically calculate the dominant force acting on the PAA bead. The BDA software has an user friendly interface (*Figure 3E*) and is available on GitHub (https://github.com/tobetz/ElasticBeadAnalysis [copy archived at swh:1:rev:96bf2d2cfc85a-ca137e39adccf6c840eb851ab72], *Wallmeyer, 2020*).

## Local tension within in vitro skeletal muscle tissues during formation

We then pursued a quantitative study of local tension within 3D skeletal muscle microtissues, which was made possible by integrating PAA beads within tissues and conducted high-resolution microscopy in our custom culture platform. Specifically, 3D skeletal muscle tissues were raised from C2C12 cells in the PMMA-based chambers in which PAA beads were included in the starting cell-matrix suspension before seeding it into the chamber to produce microtissues with PAA beads stably embedded throughout. The tissues containing the PAA beads were then monitored and the tension was

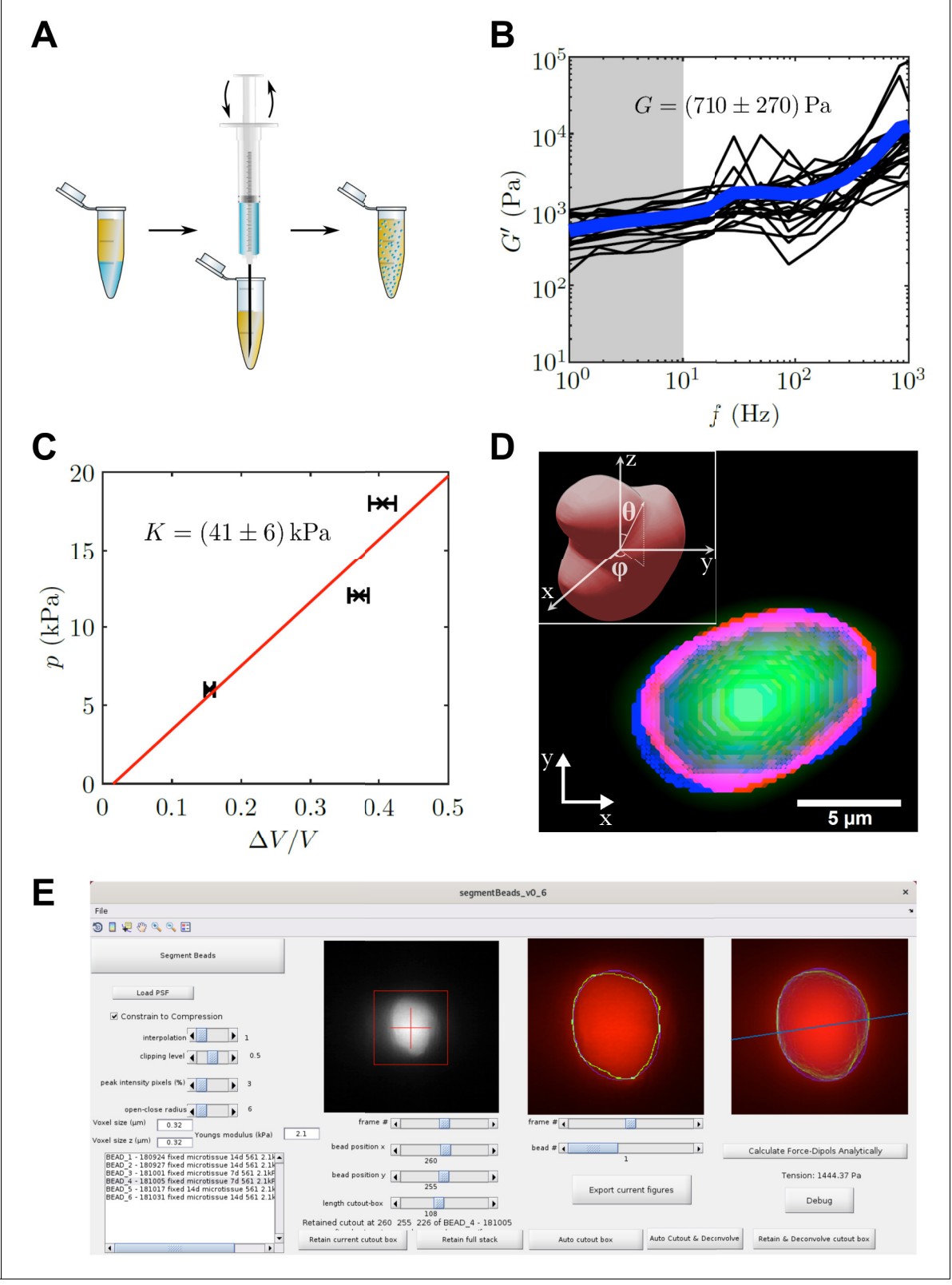

**Figure 3.** PAA bead characterization and analysis software. (**A**) Schematic workflow of bead fabrication. Using a syringe, a water in oil emulsion is created and the acryl amide solution is polymerized to produce elastic PAA beads. (**B**) Graph showing the determination of the shear modulus of elastic PAA beads, which was measured for varying oscillation-frequency of 1 μm beads. (**C**) Graph depicting the determination of the bulk modulus of elastic PAA beads, which was measured by an osmotic pressure approach using dextran. (**D**) Segmentation of a deformed PAA bead (green). The surface (red)

*Figure 3 continued on next page*

*Figure 3 continued*

was segmented and a linear combination of spherical harmonics of degree n = 0, two and order m = 0 (blue) were fitted to the surface. Insert is showing the coordinate system and angles $\theta$ and $\phi$ used for the spherical harmonics expansion. (E) Screenshot of PAA bead deformation analysis software front-end. Providing a point spread function enables the automatized program to deconvolve the image and determine the edge position of an elastic bead (green) in a given cutout box. The bead's shape is then fitted by spherical harmonics (blue), which are used to calculate the main axis of the force dipole (blue line) and the corresponding tension analytically. For detailed information also refer to section 'BDA software workflow'.

The online version of this article includes the following source data for figure 3:

**Source data 1.** Source data of shear modulus determination of PAA beads (*Figure 3B*).
**Source data 2.** Source data of bulk modulus determination of PAA beads (*Figure 3C*).

determined within the developing tissues over 2 weeks. Immunostaining of 14-day-old tissues revealed normal in vitro muscle tissue formation featuring multinucleated myotubes aligned parallel to each other with typical sarcomere structures (*Figure 4A*). Images of muscle tissue cross sections revealed dense biomimetic muscle tissues with PAA beads stably incorporated and spread evenly throughout the whole tissue (*Figure 4B*). Notably, all monitored PAA beads were located between forming myotubes (*Figure 4A*, 45° angled view, B, *Video 5*). In addition, we observed ordinary tissue dynamics and nuclear migration within myotubes in closest proximity to elastic beads (*Video 6*). These observations suggest that the presence of elastic beads does not disturb the formation of the muscle tissues.

We first imaged the PAA beads contained within the living muscle microtissues via spinning disc microscopy on the day of cell seeding. The culture chamber is crucial to obtain fluorescence images of the beads that have sufficient quality for subsequent BDA. We repeated this tension investigation in living tissues at 7 and 14 days of differentiation, respectively. Although the beads were fairly spherical directly after seeding, we observed significantly higher deformation, and therefore tension, within the tissues after one week of differentiation (*Figure 4C*). The average internal tissue tension at this time-point was slightly more than 2.4 ± 0.9 kPa, and maintained a similar level of tension following 2 weeks of differentiation. As expected, we observed that the main axis of local force dipoles was perpendicular to the myotube axis 7 days after differentiation (*Figure 4F*), which indicates that the bead deformation was due to myotube formation and remodeling. By contrast, we observed more randomly distributed angles of PAA bead deformation and therefore tension at the day of seeding. Knowing the local positions of beads within a tissue we were able to additionally display a tension map of C2C12 microtissues at the day of seeding and 7 days post differentiation. We found that the tension was homogeneously distributed throughout different regions in the muscle tissue (*Figure 4G*). As a control experiment, we measured the tension of beads embedded in ECM matrix only made of geltrex and fibrin, and without cells. We found no significant difference between beads in ECM matrix without cells after 7 days when compared to beads measured directly after seeding (*Figure 4C*). This demonstrates that the observed bead deformation is indeed due to remodeling and differentiation of the cells in the ECM matrix. To further control these measurements, we cultivated 3D muscle tissues with very stiff 150 kPa beads integrated throughout. At this high stiffness, we expect that the measured forces cannot deform the beads sufficiently for detection using microscopy. Indeed, in this case we did not observe deformation of the PAA beads after 1 week of differentiation as compared to the day of tissue seeding (*Figure 4—figure supplement 1*). Consistently, when 7-day-old tissues with PAA beads embedded are dissolved using 5% SDS, the beads return to their initial spherical shape (*Figure 4C*), demonstrating the elastic nature of the PAA beads.

To additionally validate our results acquired via spinning disk microscopy in PMMA/glass molds, we produced C2C12-derived muscle microtissues in the previously described MyoTACTIC PDMS mold (*Afshar et al., 2020*) and imaged fixed tissues that were removed from the mold and imaged in a FEP capillary using Multiview Light-Sheet microscopy to provide a comparison. Multiview Light-Sheet microscopy has the advantage of isotropic resolution, and hence checks potential problems due to spinning disk microscopy and deconvolution. We first conducted studies and confirmed that fixation of in vitro skeletal muscle tissues per se did not impair embedded PAA bead shape significantly by comparing images of the same bead before and after fixation in our PMMA/glass molds (*Figure 4—figure supplement 2*). Consistent with bead data obtained using spinning disk confocal microscopy in our PMMA/glass device, we observed a significant increase of local tension exerted

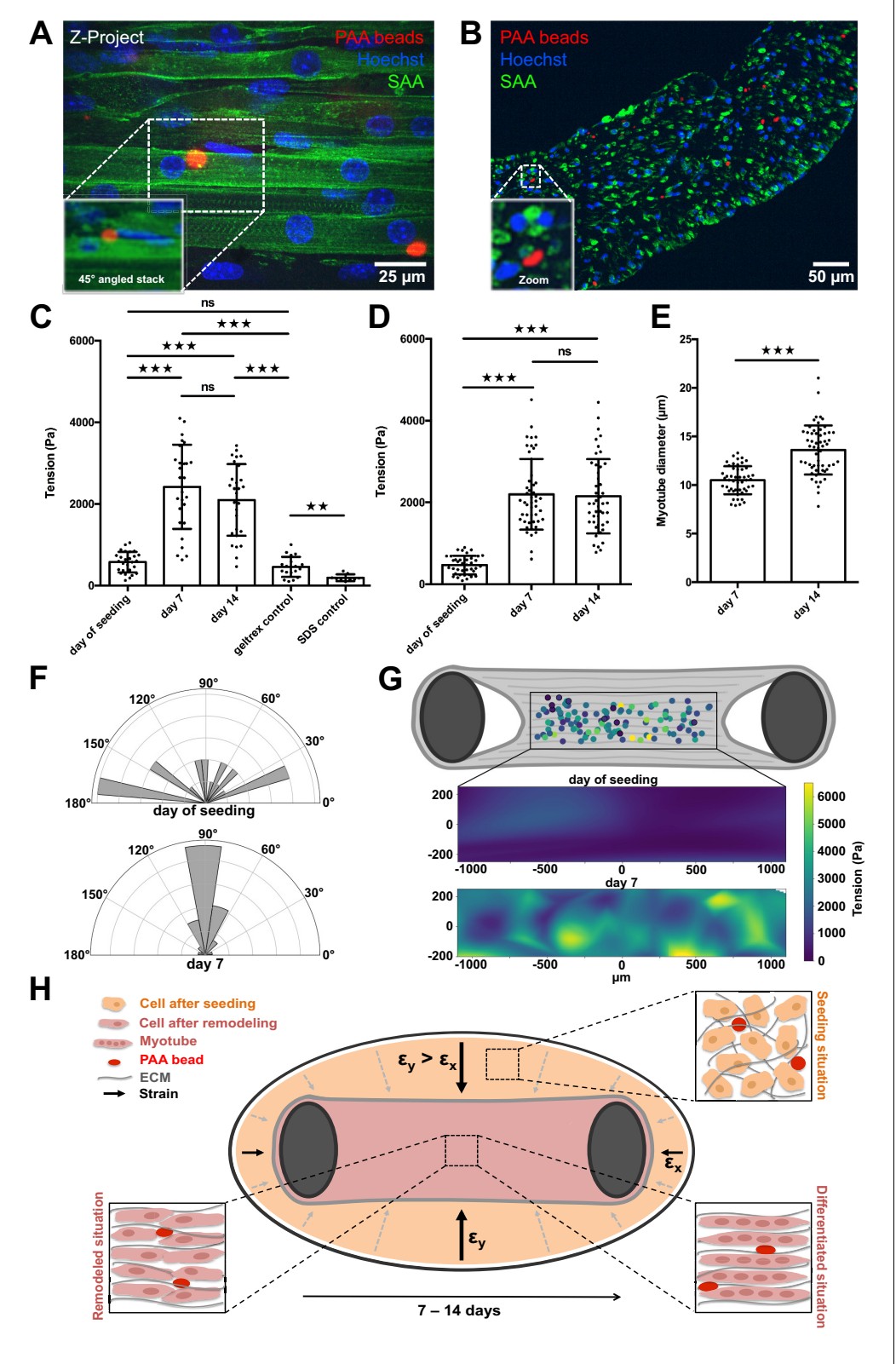

**Figure 4.** PAA beads serve as local tension sensors within in vitro muscle tissues (**A**). Representative flattened confocal microscopy stack of a 14-day-old C2C12 tissue with PAA beads embedded (red) and immunostained for sacomeric alpha-actinin (green) and counterstained with Hoechst 33342 to visualize nuclei (blue). PAA bead section is depicted as a 45° angled view of the stack to show exact inter-myotube bead localization. (**B**) Representative confocal snap of a transverse section of a 14-day-old C2C12 tissue. A PAA bead section is depicted as a zoom to clarify inter-myotube bead

*Figure 4 continued on next page*

*Figure 4 continued*

localization. (C) Bar graph of local tension within C2C12 muscle tissues at the day of seeding (N = 7, n = 28), and after 1 (N = 4, n = 28) and 2 weeks of differentiation (N = 4, n = 29). Beads were imaged in the context of living tissues raised in PMMA molds for BDA utilizing spinning disk microscopy. Also shown is negative controls in which tension was measured in the cell-free geltrex/fibrin matrix alone (N = 3, n = 20) as well as following the dissolution of tissues using 5% SDS (N = 3, n = 10). (D) Bar graph of local tension within C2C12 muscle tissues raised in the MyoTACTIC PDMS microtissue platform that were fixed and removed for imaging using Multiview Light-Sheet microscopy on the day of seeding (N = 13, n = 39), and after 1 (N = 17, n = 48) and 2 weeks of differentiation (N = 19, n = 46). (E) Bar graph of myotube diameter analysis conducted on C2C12 muscle tissues at 1 (N = 3, n = 49) and 2 (N = 3, n = 57) weeks of differentiation. Each myotube diameter data point reflects the average of three measurements per myotube. (F) Angles of PAA bead deformation at the day of seeding and 7 days after differentiation. (G) Flattened tension map of C2C12 muscle tissues at the day of seeding and 7 days after differentiation generated from BDA output. (H) Model of tension-driven in vitro muscle tissue formation. The online version of this article includes the following source data and figure supplement(s) for figure 4:

**Source data 1.** Source data of local tissue tension studies (*Figure 4C–E*, *Figure 4—figure supplements 1* and *2*).
**Source data 2.** Source data of bead deformation angles and tissue tension maps (*Figure 4F&G*).
**Figure supplement 1.** Stiff 150 kPA PAA beads do not get significantly deformed during biomimetic muscle tissue formation.
**Figure supplement 2.** Microtissue fixation with 4% PFA for 15 min does not impair PAA tension sensor outcome.

on the elastic beads after 7 days of skeletal muscle microtissue differentiation of slightly more than 2.3 ± 0.9 kPa (*Figure 4D*), but again, no significant difference was detected after a second week of differentiation. Importantly, myotube diameter increased significantly from 10.5 ± 1.4 μm at 1 week of differentiation to 13.6 ± 2.5 μm in the second week (*Figure 4E*). This shows that while the initial phase of tissue compaction myotube differentiation influences bead tension, this further increase in myotube diameter does not result in more tension on embedded PAA beads.

## Discussion

The emergent importance of mechanobiology has demonstrated that mechanical interaction between cells as well as stiffness and tensile forces provide important signaling elements for cell biology and cell fate. Muscle tissue is intrinsically exposed to large forces and rapid changes in tension and stiffness. Hence, it is to be expected that the mechanical properties of the environment are of particular relevance for the homeostasis of muscle tissue. To study these interactions, not only the precise observation of self organized 3D tissue is required, but also a non-invasive readout of global and local forces is important.

Furthermore, in vitro skeletal muscle tissue model systems provide an enormous potential to give new insights into muscle formation, degradation, repair, and dynamics. However, the usability of these novel systems has been limited by several key weaknesses of the currently used culturing methods. While the PDMS-based elastomers used in such post-based chambers is well studied and under control, its optical properties prevent the usage of high numerical aperture objectives and hence high- as well as super-resolution fluorescence microscopy. Furthermore, PDMS acts like a protein sponge that absorbs large amounts of proteins from the medium, thus preventing the use of special serum free media that are required in modern stem cell approaches. Here, we solve both problems by using an inverted geometry chamber that is made of PMMA. The material prevents the protein sponge effect, while the geometry allows positioning a glass coverslip in close proximity to the self-organized muscle tissue, thus enabling high- and even super-resolution microscopy.

Changing the material and adapting the geometry to still enable global force measurements while gaining high-quality optical access is a key advancement. It now allows time resolved high-resolution measurements of in vitro muscle tissues formation, myoblast fusion and myotube maturation, without the need of fixation and tissue removal as necessary in the previous designs. Using fluorescent protein tagging it was directly possible to gain high-resolution 3D images of actin networks with a high time resolution. Such fluorescent protein based life stains in combination with this approach will help understanding the dynamic interactions between muscle cells during differentiation, fusion, myotube formation, and maturation. Therefore, we enable real-time high-resolution imaging during cultivation of in vitro muscle tissues for the first time which is of great benefit for various future research issues such as myoblast fusion or myotube maturation. A previous 2.5D myotube culture approach was indeed able to monitor first myotube maturation and nuclear movement to the periphery of skeletal myotubes by dynamic high-resolution imaging (*Roman et al., 2017*).

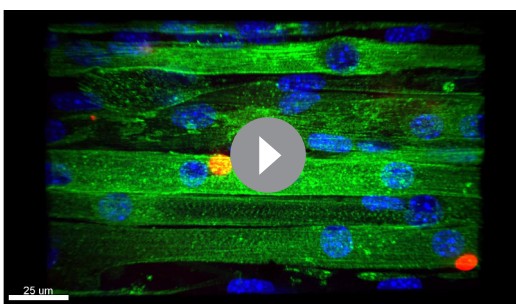

**Video 5.** 3D confocal stack rotation of a 14-day-old C2C12 tissue with PAA beads embedded (red) and immunostained for sacomeric alpha-actinin (SAA, green) and counterstained with Hoechst 33342 to visualize nuclei (blue).

https://elifesciences.org/articles/60145#video5

However, this approach cannot draw conclusions within a 3D system, is not capable to conduct functional or contraction studies and is furthermore unable to quantify force generation of the tissue, which all can be combined using our novel approach.

The here presented 3D in vitro muscle tissues raised close to a glass coverslip displayed myotube structures characteristic for progressing maturation for example sacomere striations and multinucleation (*Figure 1*) and indeed showed functional responses to contractile stimuli (*Figure 2*, *Video 2*). While the force measurement via post-deflection is a commonly used method (*Legant et al., 2009*), we now offer an advanced and reliable readout for global force generation which is even more precise due to sharply milled post edges and higher quality of imaging through glass. We therefore pave the way for future contraction studies of diseased or individual patient-related tissues. As for instance, by shortening the height of the PMMA posts we can vary the spring constant of the posts dramatically and therefore enable first isometric contraction investigations of skeletal muscle tissues in vitro which is impossible using highly flexible PDMS.

To enable the study of potential correlations of contractile forces and tension between cells, we exploit the post-deformations to determine the overall tissue contractility while the deformable beads are used to measure the local tension between individual cells and tubes. PAA beads were just previously reported to function as tension sensors within cancer spheroids in vitro, during phagocytosis as well as within zebrafish embryos in vivo (*Dolega et al., 2017*; *Lee et al., 2019*; *Träber et al., 2019*; *Vorselen et al., 2020*). Combining local and global analysis is only possible with the new chamber design, as the bead deformation analysis relies on high-resolution images. Furthermore, in contrast to the previous published tension analysis programs, the here introduced approach focuses on the principal components of tissue tension and directionality, which largely simplifies the analysis and the comparison between different treatments, cell types and contraction trigger approaches. The access to the global and the local tension also allows to independently crosscheck the values. Using the measured global tensional force as determined by the post-deflection of $f_t = 0.3 \pm 0.1$ mN (*Figure 2*) as well as the cross-section area of $A = 0.17 \pm 0.03$ mm$^2$ (*Figure 1*) of the in vitro skeletal muscle tissue, we use $t = f_t/A$ to predict a local average tension of $t = 1.8 \pm 0.67$ kPa, which is in excellent agreement to the average value of $2.4 \pm 0.9$ kPa obtained by the elastic bead analysis (*Figure 4*, see Materials and methods for details). As the bead analysis is sensitively depending on a good knowledge of the Poisson's ratio, this test provides important confirmation that the introduced measurements are reliable. Furthermore, this crosscheck suggests that the post-deflection measurements are already sufficient to determine average tension in the tissue, but the local measurement can be used to then test the tension as function of position in the tissue. Again, this conclusion is directly supported by the force maps inferred from the distribution of beads in the tissue, which did not show any obvious spatial patterns of the forces across the tissue (*Figure 4G*).

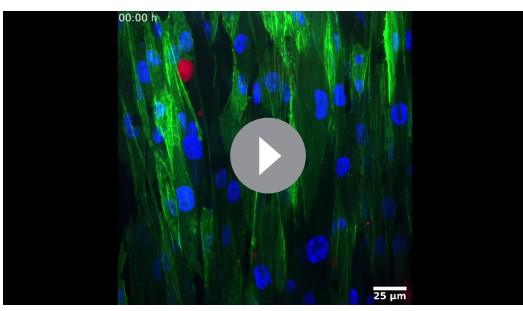

**Video 6.** Flattened 12.40 hr timeseries stack of a developing human microtissue 4 days after differentiation shows highly dynamic nuclear motion in closest proximity to a deformed PAA bead (red). Lifeact-GFP (green) and H2B-mCherry (blue) was stably introduced into AB1167 cells. Video was recorded with a ×60 water immersion objective.

https://elifesciences.org/articles/60145#video6

Establishing the combined approach for the first time in forming 3D skeletal muscle tissues in vitro, we observe a significant increase in local tension between myotubes in C2C12 cells after 1 week of differentiation which does not further increase in the following week (*Figure 4C,D*). Interestingly, we do not observe an increase of global tissue pre-tension in the second week of differentiation, either (*Figure 2*). However, we indeed monitor a significant increase in myotube diameter from week 1 to week 2 of differentiation as others reported before (*Madden et al., 2015*; *Afshar Bakooshli et al., 2019*; *Afshar et al., 2020*). In addition, Afshar et al. showed slight in vitro muscle tissue remodeling from week 1 to week 2 of differentiation. Hence, neither myotube diameter nor progressing tissue remodeling has a significant impact on local cellular tension within the 3D muscle tissues in the second week of differentiation. Therefore, a mechanical homeostasis may be reached on a cellular level among myoblast fusion, myotube death, and myotube progressing maturation after 1 week of in vitro muscle tissue differentiation. Furthermore, we can speculate that myotubes may be protected by a mechanical buffer layer composed of for example large glycocalyx biopolymers that are known to have crucial mechanical functions for cells and cell aggregates (*Gandhi et al., 2019*). Our results suggest that cellular tension within in vitro muscle tissues is more connected to tissue pre-tension although the direction of the force dipoles suggests that remodeling of the tissue and myotube formation is contributing to bead deformation.

These results can be integrated in a simple model for myoblast and myotube self organization that is dependent on tension (*Figure 4H*). As the cells and the ECM are initially seeded homogeneously, and with isotropic orientation, we observe a mechanical and structural symmetry breaking along the post directions. As the boundary condition of the chamber wall are non-adhesive, the only mechanical support is given by the anchoring of the tissue around the posts. It is well known that cells in general, and myoblasts in particular generate contractile forces on their environment. Due to the free boundary on the chamber wall these lead to an anisotropic deformation of the tissue which undergoes drastic shrinking in the direction normal to the post axis, while the shrinking is only minimal in the direction of the posts. Hence, any longitudinal object (cells and ECM fibers) will start to rotate due to the contraction, thus aligning along the post axis. This already leads to the observed alignment of the post forces, the direction measured by the beads deformation analysis and the cell alignment (*Figure 4*). Additionally, the mechanical tension in the tissue further results in a systematic pulling on the cells. From a simple viscoelastic material point of view, these forces lead to an elongation of the cells along the post axis. In such a simple model, the outcome is hence due to an initial symmetry breaking induced by the boundary conditions, which is later further enhanced by the mechanical tension that is supporting elongation in the post-direction and contraction in the direction perpendicular. Further active contributions in the cells may or may not be also triggered by the mechanics. Such active cellular reaction to forces is a highly relevant research field that can be now addressed using our chamber system.

The novel culture mold is milled using PMMA and hence, easy and reliable to fabricate. The chamber design also allows scaling to a large number of experiments in parallel, which is a key element for screening approaches. It was designed for sufficient gas exchange, easy medium exchange as well as drug delivery via holes in the lid of the mold. Utilizing PMMA and microscopy glass for our molds, we overcome the issues of immense chemical absorption by the material and poor optical properties that previous approaches possessed (*Madden et al., 2015*; *Afshar Bakooshli et al., 2019*; *Afshar et al., 2020*). Thus, we can additionally offer serum-free culturing of in vitro skeletal muscle tissues and accurate drug dose response studies.

In conclusion, we provide a novel technique for functional 3D in vitro skeletal muscle tissue cultivation that enables real-time high-resolution microscopy of living 3D biomimetic muscle tissue for the first time. We use the new approach for first global as well as local cellular force investigations of developing in vitro muscle tissues. We observe that cellular tension in C2C12 in vitro muscle tissues is closely related to global tissue pre-tension and reaches a mechanical homeostatic phase after 1 week of differentiation. However, in vitro skeletal muscle tissue maturation is still progressing in the second week of differentiation, interestingly. Further, cultivation of muscle tissues directly on glass enables whole new opportunities to study highly complex and dynamic issues of myogenesis in 3D in vitro. In addition, an easy and reliable readout of contractile forces makes the novel culture mold applicable for individualized drug screening as well as diagnosis.

# Materials and methods

## Key resources table

| Reagent type (species) or resource | Designation | Source or reference | Identifiers | Additional information |
|---|---|---|---|---|
| Strain, background (*Escherichia coli*) | Stbl3 | Invitrogen | Cat# C737303 | Chemically competent cells |
| Cell line (*M. musculus*) | C2C12 myoblast | ATCC | Cat# CRL-1772 RRID:CVCL_0188 | |
| Cell line (*Homo sapiens*) | AB1167 myoblast | Vincent Mouly *Mamchaoui et al., 2011* | | |
| Cell line (*Homo sapiens*) | AB1190 myoblast | Vincent Mouly *Mamchaoui et al., 2011* | | |
| Plasmid | Fubi-ChR2-GFP | Edward Boyden | RRID:Addgene#22051 | |
| Plasmid | pLenti6-H2B- mCherry | Torsten Wittmann | RRID:Addgene#89766 | |
| Plasmid | pLenti.PGK. Lifeact-GFP.W | Rusty Lansford | RRID:Addgene#51010 | |
| Antibody | Anti-sarcomericalpha actinin (monoclonal mouse) | Abcam | Cat# ab9465 RRID:AB_307264 | IF 1:100 |
| Antibody | anti-mouse IgG1 (polyclonal goat) | Invitrogen | Cat# A21121 RRID:AB_2535764 | IF 1:1000 |
| Other | Hoechst 33342 | ThermoFisher | Cat# 62249 | 20 µM |
| Software | Elastic Bead Analysis | This paper | *Wallmeyer, 2020*; *Hofemeier, 2021* | Software to analyse tension exerted on elastic beads using z-stack images of deformed beads |
| Software | Post-Deflection Analysis | This paper | https://github.com/Tillmuen09/PostDeflectionAnalysis | Software to analyse forces exerted on post using timelapse movies of post deflections |

## PMMA mold fabrication for 3D muscle tissue culture

Muscle tissue culture molds consists of two main parts, both milled from polymethyl methacrylate (PMMA) as depicted in *Figure 1A*. The bottom part of the PMMA chamber contains the ellipsoid culture wells (diameters: 3.5 mm, 6.5 mm) and is glued onto a microscopy cover glass (VWR, Radnor, USA) using PDMS (Sylgard 170 silicone, Sigma, St. Louis, USA) which is cured for 24 hr at room temperature. The upper part represents the PMMA lid, which extends two 16 mm long ellipsoid posts per well (diameters: 0.68 mm, 1.3 mm) with a distance of 3.3 mm between each opposing posts. Prior to use, the molds were sterilized using 70% ethanol and the wells were coated with a Poloxamere solution in ddH$_2$O over night at 4°C (5% Pluronic F-127, Sigma, St. Louis, USA) to render the surface non-adhesive. MyoTACTIC PDMS molds were fabricated exactly as described in *Afshar et al., 2020*.

## 2D and 3D skeletal muscle progenitor culture

The C2C12 mouse muscle progenitor cell line was obtained from ATCC, and authenticated by its phenotype. The AB1190 and AB1167 immortalized human muscle progenitor cell line (*Mamchaoui et al., 2011*) was obtained from Vincent Mouly (Paris, France). All cells are confirmed negative for mycoplasma contamination using PCR tests. Channelrhodopsin-2 was stably introduced into the AB1190 cell line by lentiviral transduction as described on the (*Broad-Institute, 2020*) webpage. The Fubi-ChR2-GFP plasmid was a gift from Edward Boyden (Addgene plasmid #22051; *Boyden et al., 2005*). LifeAct-GFP and H2B-mCherry was stably introduced into the AB1167 cell line by lentiviral transduction. The pLenti6-H2B-mCherry plasmid was a gift from Torsten Wittmann (Addgene plasmid #89766) (*Pemble et al., 2017*) and the pLenti.PGK.Lifeact-GFP.W plasmid was a gift from Rusty Lansford (Addgene plasmid #51010). Following lentiviral infection and a period of

culture to expand the population, the cell populations were sorted to enrich for transduced cells. C2C12 cells were cultivated in tissue culture flasks (75 cm$^2$, Greiner, Kremsmünster, Austria) in 20 ml Dulbecco's Modified Eagle Medium (DMEM, Capricorn, Ebsdorfergrund, Germany) containing 10% fetal calf serum (FCS, Sigma, St. Louis, USA) and 1% penicillin-streptomycin (Gibco, Waltham, USA) at 37°C, 5% $CO_2$ in a humidified incubator. For cultivation of AB1190 and AB1167 cells, cultivation conditions were similar to those for C2C12, with the exception that Skeletal Muscle Cell Growth medium (PROMOCELL, Heidelberg, Germany) was used as the base medium instead of DMEM.

3D skeletal muscle tissues were raised in culture as previously reported (*Madden et al., 2015*; *Afshar Bakooshli et al., 2019*). Briefly, $1.5 \times 10^7$ cells/ml were resuspended in an ECM mixture containing DMEM (40 % v/v), 4 mg/ml bovine fibrinogen (Sigma, St. Louis, USA) in 0.9% (w/v) NaCl solution in water and Geltrex (20 % v/v, Gibco, Waltham, USA). Custom-made fluorescent PAA beads were directly added to the ECM mixture (1000 beads/ml). A total of 25 µl of the cell-ECM mixture was utilized for each tissue in the PMMA molds and 15 µl for the MyoTACTIC PDMS molds (*Afshar et al., 2020*). Fibrin polymerization was induced with thrombin (Sigma, St. Louis, USA) at 0.5 units per mg of fibrinogen for 5 min at 37°C. Subsequently, 300 µl growth medium consisting of DMEM supplemented with 20% FCS, 1% penicillin-streptomycin and 1.5 mg/ml 6-aminocaproic acid (ACA, Sigma, St. Louis, USA) was added. After 2 days, the growth medium was exchanged to differentiation medium consisting of DMEM supplemented with 2% horse serum (HS, Sigma, St. Louis, USA), 1% penicillin-streptomycin (Gibco, Waltham, USA) and 2 mg/ml ACA. For human microtissues, the differentiation medium was additionally supplemented with 10 µg/ml insulin. The differentiation medium was changed every other day. For visualization of seeding and cultivation of 3D skeletal muscle tissues also refer to *Video 7*.

## PAA bead fabrication

To produce PAA beads a water-in-oil emulsion approach was used. Since in this study predominantly 2.1 kPa (Young's modulus) beads were utilized, the fabrication and characterization for this elasticity is described, representatively. To generate beads with other elasticity, adapted concentrations of monomers and crosslinkers are required.

The water phase was prepared by mixing two parts of acrylamide solution (40 % v/v, Sigma, St. Louis, USA) with one part of N,N'-methylenebisacrylamide solution (2 % v/v, Sigma, St. Louis, USA). This PAA solution was then diluted to 13.5% (v/v) using a 65% (v/v) phosphate buffered saline (PBS, Sigma, St. Louis, USA) to obtain a pre-bead-mix. The mechanical properties of the PAA beads can be tuned by changing the dilution of the pre-bead-mix. The oil phase was composed of 3% Span80 (Sigma, St. Louis, USA) in n-hexane (Merck, Darmstadt, Germany). Shortly before polymerization the pre-bead-mix as well as the oil phase were degassed for 10 min at 50 mbar. The free-radical cross-linking polymerization of the PAA solution was initiated by adding 1.5% (w/v) ammonium persulphate (APS, AppliChem, Darmstadt, Germany) and the pH-value of the solution was neutralized using NaOH solution. The emulsion was then generated by injecting the pre-bead-mix into the n-hexane with a 100 µl Hamilton syringe (Hamilton, Reno, USA). Hereafter, the polymerization was catalysed by adding 3% (v/v) N,N,N',N'-tetramethylethylenediamine (TEMED, Sigma, St. Louis, USA) and the emulsion was degassed again for 6 min. The supernatant was discarded and the polymerization was kept at 85°C for 10 min. To reach end of gelation the beads were incubated at room temperature over night. Next day, the beads were washed five times with n-hexane and transferred to 65% PBS. Finally, the beads were labeled fluorescently with ATTO-565-NHS-ester solution (Atto-Tech, New York, USA). For that purpose, the labeling solution was incubated with the beads for at least 30 min at room temperature and washed three times with 65% PBS afterwards. The fluorescent intensity of the beads was increased by repeating the labeling process three to five times.

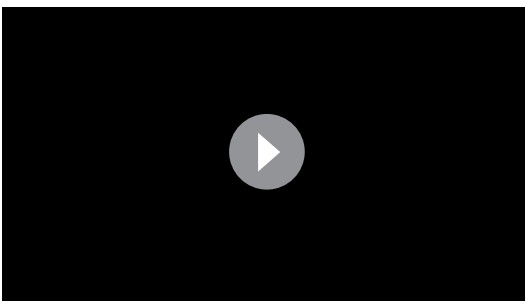

**Video 7.** Video tutorial for seeding and cultivation of biomimetic 3D skeletal muscle tissues.
https://elifesciences.org/articles/60145#video7

## Mechanical characterization of PAA beads

To characterize the mechanical properties of the elastic PAA beads the shear modulus $G$ and bulk modulus $K$ were determined. The shear modulus was measured via active microrheology using a custom optical tweezers setup. For this purpose, smaller beads with a diameter of 1 μm (Latex beads, Sigma, St. Louis, USA) were embedded into the PAA beads. The complex shear modulus, $G^* = G' + iG''$, was measured for varying oscillation-frequency of the 1 μm beads (*Ahmed et al., 2018*). The shear modulus $G$ used for further analysis was estimated by averaging shear storage moduli $G'$ for low frequencies from 1 to 10 Hz. The bulk modulus $K$ was measured via an osmotic pressure approach using 2 MDa dextran molecules (Dextran T2000, Pharmacosmos, Wiesbaden, Germany) which were dissolved in the PAA beads solution. Since the pore size of the PAA gel is much smaller than the hydrodynamic radius of the dextran, an osmotic pressure $p = K\frac{\Delta V}{V}$ acts on the PAA beads and gives rise to a compression $\Delta V/V$ proportional to the bulk modulus $K$. The exerted osmotic pressure depends on the concentration of dextran and has been calibrated previously (*Monnier et al., 2016*). Here, three different concentrations (60, 85, and 100 g/l) were used, corresponding to osmotic pressures of 6, 12, and 18 kPa, respectively. The distribution of PAA bead diameters in the three solutions was measured and compared to the uncompressed situation by segmenting images of PAA beads with the find-edges-function of Fiji and fitting ellipses to the edges. The bulk modulus $K$ was determined by a linear fit of the volume change $\Delta V/V$ to osmotic pressure $p$.

## Bead deformation analysis (BDA)

In order to reconstruct forces acting in the tissue, the deformation of embedded PAA beads were quantified using a two-step process implemented in a graphical user interface developed in Matlab (Mathworks, US). First, the beads were segmented and fitted using spherical harmonics. Then an analytical solution to the elastic problem was applied to gain the forces on the beads. For bead segmentation, a small volume around each PAA bead was cut out of the 3D image stack and if desired the effective pixel density was interpolated by a factor of 2 for display purposes. In the cutout, a threshold intensity was determined according to segmentation parameters that were typically set to *clipping level* (50%) and *peak intensity voxels* (3%) and the image was then transformed into a binary image. Subsequently, the surface was determined by extracting voxels at the edge of the binary volume and the algorithm returns the coordinates $(x_i, y_i, z_i)$ of these voxels that do form the surface of the PAA bead. In order to quantitatively capture the deformation, spherical harmonics were fitted to the segmented bead surface using a Nelder-Mead simplex algorithm. As we were only interested in the dominant forces acting on the bead, we took advantage of the mode decomposition in spherical harmonics and restricted the fit to spherical harmonics $Y_{nm}(\theta, \phi)$ of degree $n = 0, 2$ and order $m = 0$. While the $Y_{00}$ mode gives the radius of the sphere, the $Y_{20}$ mode is the mode of dominant compression, where the z-direction was determined by rotating the sphere until the absolute value of the $Y_{20}$ mode became maximal. This ansatz focuses on the direction of major deformation of the PAA bead, and it furthermore uses the angular symmetry around the z-axis that is implied in the order $m = 0$.

In detail, following this approach, the coordinates of the segmented surface $(x_i, y_i, z_i)$ were transformed into spherical coordinates $r(\theta, \phi)$. The surface defined by the spherical harmonics

$$r_{\mathrm{sh}}(\theta) = \sum_{n=0,2} c_{n0} Y_{n0}(\theta, \phi) = \frac{c_{00}}{\sqrt{4\pi}} + c_{20}\sqrt{\frac{5}{16\pi}} \cdot (3\cos^2\theta - 1)$$

was then fitted to the segmented surface by rotating $r_{\mathrm{sh}}(\theta, \phi)$ to align with the major direction of deformation of $r(\theta, \phi)$ and minimizing the residual $(r_{\mathrm{sh}} - r)^2$ by varying $c_{00}$ and $c_{20}$. Thus, a sphere of radius $c_{00}/\sqrt{4\pi}$ with uniaxial deformation proportional to $c_{20}$ was fitted to the deformed PAA bead. This fitting procedure was repeated five times with randomly picked starting values in each case. The parameter combination with the lowest residual was taken as the result. The sought force dipole $\vec{F}$ acting on the PAA bead was finally calculated by

$$\vec{F} = \frac{G}{2}c_{00}c_{20}\sqrt{5}\left(\frac{N_{\mathrm{r}}(\nu)}{4} + \frac{3N_\theta(\nu)}{2}\right)\vec{e}_{\mathrm{z}} \tag{1}$$

$$N_{\mathrm{r}}(\nu) = \frac{2\nu}{1-2\nu} + \frac{1}{2(2-3\nu)(1-2\nu)} + \frac{1}{2(2-3\nu)} \tag{2}$$

$$N_{\theta}(\nu) = \frac{1}{2} - \frac{1}{2(2-3\nu)} \tag{3}$$

with the scaling factors $N_r$ and $N_\theta$ as well as $\vec{e}_z$ being the unit vector on the main deformation axis. *Equation 1* was derived following classical elasticity theory (*Lurie and Belyaev, 2005*), resulting in expression for the stress on the sphere that was then integrated over the half sphere to get the force dipole acting on the PAA bead. The axis of the force dipole relates to the axis of rotation when fitting the spherical harmonics. Here, the term $1/(1-2\nu)$ shows that the model diverges for fully incompressible materials. Hence, for a good measurement either the Poisson's ratio must be determined very precisely, or materials with Poisson's ratio much smaller than 0.5 should be sought.

## Immunostaining and confocal fluorescence microscopy

3D skeletal muscle tissues were washed once with PBS and fixed in 4% paraformaldehyde (PFA) for 15 min at room temperature in the culture device and afterwards washed three times with PBS, again. Frozen tissues were sectioned, mounted as 12 µm sections and rehydrated. Next, the tissues were blocked for 1 hr at room temperature using PBS supplemented with 10% goat serum (GS, Sigma, St. Louis, USA) and 0.2% Triton-X-100 (Carl Roth, Kalsruhe, Germany). Subsequently, the samples were incubated with the primary antibody (monoclonal mouse anti-sarcomeric alpha actinin, 1:100, Abcam, Cambridge, UK) diluted in blocking solution over night at 4°C. After three washes with blocking solution, the samples were incubated with the appropriate secondary antibody (polyclonal goat anti-mouse IgG H and L, 1:1000, Abcam, Cambridge, UK) diluted in blocking solution for 45 min at room temperature. The cell nuclei were counterstained using Hoechst 33342 (1:1000, ThermoFisher, Waltham, USA). Confocal images were acquired within the first 100 µm of the tissue using Slidebook six software (3i, Denver, USA) using an inverted microscope (Nikon Eclipse Ti-E, Minato, Japan) equipped with a CSU-W1 spinning disk head (Yokogawa, Musashino, Japan) and a scientific CMOS camera (Prime BSI, Photometrics, Tucson, USA). Images were analysed and prepared for publication using the open source software Fiji (*Schindelin et al., 2012*).

## Super-resolution dSTORM imaging

Measurements were performed using a custom-built optical setup as previously described (*Oleksiievets et al., 2020*). Briefly, the excitiation laser (638 nm, PhoxX+ 638–150, Omicron) was coupled in the imaging objective (UAPON 100X oil, 1.49 NA, Olympus), and the sample was positioned using a two-axis linear stage (M-406, Newport). The emmited fluorescence was spectrally separated from the excitation path and imaged onto the chip of an emCCD camera (iXon Ultra 897, Andor), resulting in an effective pixel size in sample space of 103.5 nm.

In order to create blinking conditions for Alexa 647, 200 µl of enzymatic oxygen scavenging buffer (glucose oxidase 0.5 mg/ml, catalase 40 g/ml, glucose 10% w/v in PBS pH 7.4) with addition of thiol (50 mM Cysteamine) was injected into the sample chamber. A thin FEP tube (diameter 2.4 mm) was introduced into the sample chamber and positioned on top of the muscle tissue to further bring the tissue into closest proximity to the coverslip surface for ×100 image acquisition. Image stacks of 40,000 frames were acquired with an exposure time of 30 ms. The recorded images were then analyzed with an ImageJ plugin (ThunderSTORM; *Ovesný et al., 2014*) for determining the positions of single emitters. All images were corrected for lateral drift, and out-of-focus localization was rejected based on the width of the single-molecule images. The average localization precision of the reconstructed super-resolution image was 17 nm. A Fourier Ring Correlation (FRC) map was created using the NanoJ-SQUIRREL plugin (*Culley et al., 2018*), which allowed us to determine the average resolution of 65 nm, with a minimum value of 33 nm.

## Multiview imaging

Fixed muscle tissues were mounted in fluorethylenpropylene (FEP) capillaries (Proliquid, Überlingen, Germany) together with 1% low-melting-agarose solution (LMA, Invitrogen, Carlsbad, USA) for multiview imaging using a Z1 Light-Sheet microscope (Zeiss, Oberkochen, Germany). The LMA solution

contained fluorescent sub-resolution beads (100 nm diameter, Invitrogen, Carlsbad, USA) that were used for registration of the images acquired from different views. The samples were imaged from at least four different views with angles in the range 60–90° between the views. Afterwards, the images from different views were registered and fused using the Multiview Deconvolution software (*Preibisch et al., 2014*). The program detects the positions of the registration beads and overlays the images accordingly. The registration beads were also used to estimate the point spread function (PSF) of the microscope. Knowing the PSF, the images were deconvolved and fused.

### Post-deflection analysis

For post-deflection analysis, we focused the top part of the post with the imaging system and recorded a time series. Thanks to the high-resolution microscopy compatibility of the new chamber design, the outline of the post showed a strong contrast to the surrounding tissue. Utilizing a custom written program in Matlab (Mathworks, Natick, USA), a line of pixels located at the center of the post of each image was chosen for every time point. By calculating the gradient of pixel intensities along this line, the outline of the post appeared as a strong peak in the signal. Subpixel resolution was achieved by the fit of a Gaussian function to these peaks and determining the central position. Subsequently, the pixel values were converted into units of length using the pre-calibrated pixel sizes. Analysis of the entire time stack in that manner gave rise to a time-dependent displacement signal for the post edges. Finally, the pulling forces were determined by multiplication of the post displacement by an apparent spring constant of the post. This apparent spring constant was calculated by finite element analysis (Inventor, Autodesk, San Rafael, USA) considering the post's exact geometry and material properties. The assumption of a linear force to displacement relation is valid for the observed displacements of less than 45 µm.

### Spring constant measurements

PMMA chamber lids with posts of 16, 12, and 8 mm length were mounted horizontally into a Micro Forge MF-830 Microscope (Narishige, Japan) and an Eppendorf tube ($m_0 = 1.033$ g) was attached to the very tip of a post. The tube was incrementally loaded with $m_1 = 100$ µg, while the post-deflection $x$ was monitored under the microscope. Consequently, the spring constant $k$ was determined according to $k = g(m_0 + m_1)/x$ where $g = 9.81 m/s^2$ is the gravitational acceleration.

### Relation between directionality of tension

As crosscheck between the global (post based) and local (bead based) tension we can estimate the expected local tension using the post-deflection measurements. First, we determine the tensional stress in the tissue along the post-post axis by dividing the global force $f_t = 0.3 \pm 0.1$ mN by the tissue diameter $A = 0.17 \pm 0.03 mm^2$, yielding a tensional stress of $t = 1.8 \pm 0.67$ kPa. To compare this tension, which acts along the length of the tissue (x-axis) with the deformation which is typically along y and z, a simple calculation shows that in isotropic materials of Poisson's ratio $\nu \approx 0.5$ the absolute stress values are the same in all directions. This can be simply demonstrated by recalling Hooke's law in 3D:

$$\sigma_{xx} = \frac{E}{(1+\nu)(1-2\nu)}((1-\nu)u_{xx} + \nu(u_{yy} + u_{zz})) \qquad (4)$$

$$\sigma_{yy} = \frac{E}{(1+\nu)(1-2\nu)}((1-\nu)u_{yy} + \nu(u_{xx} + u_{zz})) \qquad (5)$$

$$\sigma_{zz} = \frac{E}{(1+\nu)(1-2\nu)}((1-\nu)u_{zz} + \nu(u_{xx} + u_{yy})). \qquad (6)$$

and considering a constraint deformation, where only a deformation in x is considered and y,z are assumed to be fixed. This constraint deformation enforces that only the component $u_{xx}$ is different from zero, and hence Hooke's law simplifies to

$$\sigma_{xx} = \frac{E(1 - \nu)}{(1 + \nu)(1 - 2\nu)} u_{xx} \tag{7}$$

$$\sigma_{yy} = \sigma_{zz} = \frac{\nu E}{(1 + \nu)(1 - 2\nu)} u_{xx} \tag{8}$$

which means that the stresses are not independent, but that

$$\sigma_{yy} = \sigma_{xx} \frac{\nu}{1 - \nu}. \tag{9}$$

In first order, the tissue around the bead remains mostly undeformed for the small deformations measured, hence the absolute stress $\sigma_{yy}$ acts on the beads. For a Poisson's ratio $\nu \approx 0.5$ the tensional stress in x direction is equal in magnitude to the measured stress in y and z direction.

Therefore, we can directly compare the measured local tension derived by the bead deformation of $2.4 \pm 0.9$ kPa to the global tension of $1.8 \pm 0.67$ kPa, and confirm that these match within the error.

## Statistical analysis

All results are presented as mean ± standard deviation if not mentioned differently and statistical differences of experimental groups were analysed by unpaired t-test using GraphPad Prism 6.0 software, where $p<0.05$ was considered as significant. Significances were subdivided into three levels: *, $p=0.05$–$0.01$; **, $p=0.01$–$0.001$; and ***, $p<0.001$. The number of biomimetic muscle tissues (N) and analysed PAA beads (n) is indicated in every figure legend. In *Figure 4E* 'n' refers to the number of measured myotubes.

## BDA software workflow

The analysis of a dataset starts by uploading .tiff files into the app. The user can simply go to *File⸽ Open folder⸽* and select the folder in which the data is stored. All the available .tiff will be shown in the lower left exploration window of the user interface. In case a deconvolution is necessary, the user can also load the corresponding Point Spread Function clicking on *Load PSF*. It is important to remark that only one PSF for deconvolution can be used in each session. Before starting the bead selection, following inputs should be checked and adapted:

- Interpolation: the predetermined value '1' will not change the raw data. By increasing it to n, a 3D interpolation algorithm is applied that returns a .tiff $n^3$ bigger.
- Clipping level: parameter responsible for the relative intensity value at which the bead surface is defined. The predetermined value 0.5 will define the bead surface at 'half maximum' of a fitted Gaussian.
- Peak intensity pixels: related to the previous parameter, sets the percentage of brightest pixels used to define an average peak value in the intensity profile. The given 3% works usually for most data.
- Open-close radius: in case of heterogeneous fluorescence, a watershed algorithm fills in the dark regions within a bead to avoid fitting problems.
- Voxel sizes of your acquisition.
- Young modulus of the bead polymer.

By clicking on the exploration window each .tiff will be loaded on the analysis deck. Using the first column of scrolls, the position of each bead can be defined along with a cutout region around it (shown by a red rectangle). This information is saved by clicking *Retain current cutout box* or *Retain and Deconvolve cutout box* (in that case a valid PSF must have been uploaded previously). In case the .tiff is already tight enough around the bead, the function *Retain full stack* can be used.

When all beads in the dataset have been selected, the analysis continues clicking on *Segment beads*. A dialog box with a progress bar indicates then the actual operation; for example 'Deconvolving bead 4 out of 13'. Once done the user can explore the resulting fits using the second plotting canvas and its corresponding slides. At last, force dipole responsible of the bead deformation is obtained directly on the screen by clicking *Calculate Force-Dipols Analitically*. The information of all the analyzed beads will be stored automatically in the origin folder. There the user finds:

- *FileName_segmentation*: a folder containing .tiff files with the cutout bead, the filtered version and the binary surface, along with .txt files containing the Spherical Harmonics coefficients obtained in the fit and the force and tension.
- *Beads.mat*: a Matlab structure with all the previous information properly labeled and ready for further analysis or plotting.

## Code availability

The bead deformation analysis as well as the post-deflection open source software is available on GitHub (*Wallmeyer, 2020*; https://github.com/tobetz/ElasticBeadAnalysis; copy archived at (swh:1: rev:96bf2d2cfc85aca137e39adccf6c840eb851ab72) and *Muenker, 2020*; https://github.com/Till-muen09/PostDeflectionAnalysis; copy archived at (swh:1:rev:00d39f7b0e52e898a6c91353b3514a-c6a817f544), respectively).

## Acknowledgements

This work was funded by the Human Frontiers Science Program (to PMG and TB). MEA received funding from the Natural Sciences and Engineering Research Council Training Program in Organ-on-a-Chip Engineering and Entrepreneurship Scholarship. PMG is the Canada Research Chair in Endogenous Repair. Funding to PMG is from the Natural Sciences and Engineering Research Council and Medicine by Design, a Canada First Research Excellence Program. TB was supported by the European Research Council (consolidator grant number 771201) and TB and JE by the Deutsche Forschungsgemeinschaft (DFG, German Research Foundation) under Germany's Excellence Strategy - EXC 2067/1- 390729940. NO is grateful to the Deutsche Forschungsgemeinschaft (DFG) for financial support via Project A10 of the SFB 803. We express our thanks to Vincent Mouly and the Myoline platform from the Institut de Myologie (Paris, France) who kindly provided the human immortalized cell line. Further we thank Yannik Vaas for his excellent technical expertise during mold fabrication and Dr. Oleksii Nevskyi for helping with super-resolution dSTORM imaging.

## Additional information

### Competing interests

Arne D Hofemeier, Timo Betz: Patent application filed for the chamber. Application number: LU101799. The other authors declare that no competing interests exist.

### Funding

| Funder | Grant reference number | Author |
| --- | --- | --- |
| Human Frontier Science Program | RGP0018/2017 | Penney M Gilbert<br>Timo Betz |
| H2020 European Research Council | 771201 | Timo Betz |
| Natural Sciences and Engineering Research Council of Canada | RGPIN-4357 | Penney M Gilbert |
| Natural Sciences and Engineering Research Council of Canada | RGPIN-7144 | Penney M Gilbert |
| Natural Sciences and Engineering Research Council of Canada | Training Program in Organ-on-a-Chip Engineering and Entrepreneurship Scholarship | Mohammad Ebrahim Afshar |
| University of Toronto | Canada Research Chair in Endogenous Repair | Penney M Gilbert |
| Deutsche Forschungsgemeinschaft | SFB 803 | Roman Tsukanov<br>Nazar Oleksiievets<br>Jörg Enderlein |

| Deutsche Forschungsge-meinschaft | EXC 2067/1- 390729940 | Jörg Enderlein Timo Betz |

The funders had no role in study design, data collection and interpretation, or the decision to submit the work for publication.

## Author contributions

Arne D Hofemeier, Conceptualization, Data curation, Formal analysis, Investigation, Visualization, Methodology, Writing - original draft, Writing - review and editing; Tamara Limon, Formal analysis, Investigation; Till Moritz Muenker, Data curation, Software, Visualization; Bernhard Wallmeyer, Data curation, Software, Methodology; Alejandro Jurado, Software, Formal analysis, Methodology; Mohammad Ebrahim Afshar, Majid Ebrahimi, Resources, Methodology; Roman Tsukanov, Nazar Oleksiievets, Investigation, Methodology; Jörg Enderlein, Conceptualization, Funding acquisition, Methodology; Penney M Gilbert, Resources, Supervision, Validation, Methodology, Writing - review and editing; Timo Betz, Conceptualization, Resources, Software, Formal analysis, Supervision, Funding acquisition, Validation, Visualization, Project administration, Writing - review and editing

## Author ORCIDs

Jörg Enderlein (ID) https://orcid.org/0000-0001-5091-7157
Penney M Gilbert (ID) https://orcid.org/0000-0001-5509-9616
Timo Betz (ID) https://orcid.org/0000-0002-1548-0655

## Decision letter and Author response

Decision letter https://doi.org/10.7554/eLife.60145.sa1
Author response https://doi.org/10.7554/eLife.60145.sa2

## Additional files

### Supplementary files

- Transparent reporting form

## Data availability

All data generated or analysed during this study are included in the manuscript and supporting files. Programs are published on GitHub.

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
