## [Decision Letter]

**Acceptance summary:**

This paper presents a novel method of culturing 3D skeletal muscle tissues using a new type of microfabricated chamber. This allows for both high resolution imaging and non-invasive force measurements, which is not accessible with the traditional methods. The global and local tensions in biomimetic skeletal tissues can be measured, showing that the tension in the tissue remains constant after 1 or 2 weeks culture, although the myotube diameters double during this period.

**Decision letter after peer review:**

Thank you for submitting your article "Global and local tension measurements in biomimetic skeletal muscle tissues reveals early mechanical homeostasis" for consideration by *eLife*. Your article has been reviewed by three peer reviewers, including Patricia Bassereau as the Reviewing Editor and Reviewer #1, and the evaluation has been overseen by Aleksandra Walczak as the Senior Editor. The following individual involved in review of your submission has agreed to reveal their identity: Natasha Chang (Reviewer #3).

The reviewers have discussed the reviews with one another and the Reviewing Editor has drafted this decision to help you prepare a revised submission.

Summary:

This Tools and Resource manuscript presents a novel method for culturing 3D skeletal muscle tissue with both high resolution imaging and non-invasive force measurements. With the unique design of the microfabricated chamber, biomimetic muscle tissues grow in close proximity to a microscopy coverslip, which permits high resolution imaging and to capture dynamic events during myotube formation and contraction without manipulation. Using elastic PAA beads as microsensors, it is possible to measure global and local tensions in biomimetic skeletal tissues. With this method, the authors show that the tension in the tissue is the same after 1 or 2 weeks culture, although the myotube diameters double during this period. This technique is highly innovative and represents a significant improvement over traditional methods of 3D muscle culturing.

Essential revisions:

The two reviewers and myself think that your method could be in principle of strong interest in the mechanobiology field. However, we think that you should complement your current work with some additional data.

1) You should show that you can change/reduce the stiffness of the PMMA posts in a controlled manner. It should be also clarified if the stiffness of the posts has been calculated or measured.

2) The quality of the muscle images is not convincing and should be improved to demonstrate than it is better than on PDMS, in particular of the acto-myosin structures during maturation (Figure 1F). Better images of the PAA beads are also needed to understand where the beads are localized.

3) Could you check that the PAA beads do not impact nucleus motion?

4) Since the force dipole is a direct output of the method, it would be interesting to display force dipole maps and the relation with the myotube axis, over time.

5) The Poisson ratio (about 0.5) deduced from the osmotic swelling and from the micro-rheology experiments shows that the PAA gel is basically incompressible in solution, which is expected but should be mentioned. In this case, can you comment how the force dipole is calculated from Equation 1, since Nr(nu) should diverge if nu=0.5 (Equation 2)?

---

## [Author Response]

Essential revisions:The two reviewers and myself think that your method could be in principle of strong interest in the mechanobiology field. However, we think that you should complement your current work with some additional data.1) You should show that you can change/reduce the stiffness of the PMMA posts in a controlled manner. It should be also clarified if the stiffness of the posts has been calculated or measured.

This is indeed a very important point. In our initial manuscript we only simulated the apparent spring constant for 16 mm long posts using the Autodesk Inventor finite element analysis. We agree that this should be crosschecked by experimental confirmation. In our revised version we did both, simulating the spring constant of a series of post length (from 4mm to 16mm) and additionally measured the spring constant for 16mm, 12mm and 8mm long posts (Figure 2F). The measurements show that for 12mm and 16mm long posts, which were used in our study, the measured spring constant is in excellent agreement with the simulated data. For shorter posts, there starts a discrepancy between the simulations and the measurements, suggesting that here manufacturing variations become important, and the user should experimentally crosscheck the constants after generating the posts. Hence, in the revised paper we point out that on the one hand the stiffness of PMMA posts is tuneable by changing the post length and on the other hand we show that the spring constant we use in our study for global force measurements is indeed reliable.

2) The quality of the muscle images is not convincing and should be improved to demonstrate than it is better than on PDMS, in particular of the acto-myosin structures during maturation (Figure 1F). Better images of the PAA beads are also needed to understand where the beads are localized.

We agree that this is an important point of the paper, as current PDMS chambers suffer from low image quality. To better demonstrate this point, we now include a direct comparison between our chamber versus classical PDMS chambers. Figure 1E is image with the highest possible magnification (20x air) through the MyoTACTIC PDMS platform, and gives only poor image quality, while Figure 1D shows improved image quality obtained by imaging through the glass bottom of our chamber. The direct comparison shows that imaging in our chambers allows to resolve the striation, while imaging through PDMS was not able to resolve these details. Obviously, this is a consequence of the higher NA objectives that can be now used to visualize reconstituted 3D muscle tissue. Such objectives cannot be used with previous, PDMS based chambers.

As pointed out by the reviewers, imaging muscle cells using live imaging of acto-myosin structures is indeed hard to do in general. However, we would like to point out that this problem is independent of the chamber, but is a result of the stably expressing Lifeact-GFP cells. Of course, such life stains are required to obtain dynamic data on the muscle tissue formation. However, the downside is that Lifeact does not specifically stain striated structures, such as fluorescently labelled proteins associated to the z-disk would do. Although this can be seen as disadvantage, Lifeact-GFP provides important advantages as it allows to monitor the overall actin structures during the full time of muscle tissue formation. As we are primarily interested in mechanical changes within the muscle during tissue formation, access to tissue dynamics is most interesting for us. Therefore, we had generated a cell line stably expressing Lifeact-GFP. To demonstrate the overall cellular dynamics we show hence the Lifeact-GFP stain, despite the lower image quality resulting from the staining method.

To address the reviewers concerns regarding the life imaging, we improved the imaging quality for Figure 1G by recording another video of a region which exhibits myoblast fusion events during muscle tissue formation. These dynamics are very well observable using the Lifeact-GFP stain. Moreover, to demonstrate that our chamber is compatible with recent high-end microscopy techniques, we imaged myofibrillar structures using super resolution dSTORM microscopy with a 100x high end objective (Figure 1H). Recording such images in the post system has never been achieved with the PDMS system. Hence, imaging through the glass bottom of our chamber system is now expanding the possibilities for investigating muscle tissue formation at super-resolution.

To further clarify the localisation of PAA beads between myotubes we included an angled view of the 3D stack image in Figure 4A and additionally we presented a rotation video of the stack in Video 5. Furthermore, we pointed out the bead location by zooming into the cross-section image in Figure 4B.

3) Could you check that the PAA beads do not impact nucleus motion?

This is indeed a very important question. We checked the impact of PAA beads on nuclear motion by acquiring videos of forming biomimetic muscle tissues with PAA beads included and stably expressing Lifeact-GFP and H2B-mCherry. We can now show that PAA beads did not reveal to have any impact on nuclear motion since nuclei were moving along PAA beads without any disturbances (Video 6). This data and a short discussion are now included in the revised version.

4) Since the force dipole is a direct output of the method, it would be interesting to display force dipole maps and the relation with the myotube axis, over time.

We thank the reviewers for this interesting remark. We indeed were able to use the output of BDA to create a tension map of C2C12 tissues on the day of seeding and 7 days after differentiation (Figure 4G) and additionally, we compared the directionality of the force dipoles on these days (Figure 4F). As expected, no alignment and only small forces are observed on the day of seeding (Figure 4F, top), but the force increased and the dipoles were aligned perpendicular in relation to the myotube axis 7 days after differentiation (Figure 4F, bottom). Please note that the forces pushing the beads are not expected to be in the direction of the muscle cell, but that the cells push the beads normal to their elongation axis, which is directly what we see. Furthermore, we do not see any clear sign of spatial differences of the forces within the observed regions.

5) The Poisson ratio (about 0.5) deduced from the osmotic swelling and from the micro-rheology experiments shows that the PAA gel is basically incompressible in solution, which is expected but should be mentioned. In this case, can you comment how the force dipole is calculated from Equation 1, since Nr(nu) should diverge if nu=0.5 (Equation 2)?

Again, this is an important point raised by the reviewers. In fact, there is no real solid material that is perfectly incompressible, and consequently most elasticity models collapse when injecting a Poisson’s ratio of exactly 𝜈𝜈=0.5. This can be for example demonstrated by looking at the dependence between the shear modulus G and the compression modulus K, which reads 𝐺𝐺=3 𝐾𝐾1−2𝜈𝜈2(1+𝜈𝜈). As every real material has a finite compression modulus, a Poisson’ ratio of exactly 0.5 would directly imply a zero shear modulus, which is just one of the hallmarks of a fluid.

Hence, there we know of no way to mathematically consistently perform these calculations with a Poisson’s ratio of exactly 0.5 for a solid material. This can also be seen when trying to model the deformation by finite element approaches. There the FEM algorithm do typically not converge when using 0.5 as Poisson’s ratio.

The high sensitivity on the Poisson’s ratio when approaching 0.5 is in fact the reason why we performed two measurement of the mechanical properties, instead of simply using a literature value for the Poisson’s ratio. Additionally, we carefully crosscheck the final results when comparing the average tension obtained globally from the post deflection with the mean measured local tension obtained from the bead deformations. As this crosscheck gives consistent values we are convinced that the presented approach is correct. However, it might be important to point out explicitly that the analytical approach is not possible for idealized incompressible materials. We now explicitly mention this point, and explain the importance of a precise measure of the Poisson’s ratio for the analysis.